# All-optical control of lead halide perovskite microlasers

Nan Zhang[1], Yubin Fan [1], Kaiyang Wang[1], Zhiyuan Gu[1], Yuhan Wang[1], Li Ge [2,3], Shumin Xiao[1,4] & Qinghai Song [1,4]

Lead halide perovskites based microlasers have recently shown their potential in nanophotonics. However, up to now, all of the perovskite microlasers are static and cannot be dynamically tuned in use. Herein, we demonstrate a robust mechanism to realize the all-optical control of perovskite microlasers. In lead halide perovskite microrods, deterministic mode switching takes place as the external excitation is increased: the onset of a new lasing mode switches off the initial one via a negative power slope, while the main laser characteristics are well kept. This mode switching is reversible with the excitation and has been explained via cross-gain saturation. The modal interaction induced mode switching does not rely on sophisticated cavity designs and is generic in a series of microlasers. The switching time is faster than 70 ps, extending perovskite microlasers to previously inaccessible areas, e.g., optical memory, flip-flop, and ultrafast switches etc.

[1] State Key Laboratory on Tunable laser Technology, Ministry of Industry and Information Technology Key Lab of Micro-Nano Optoelectronic Information System, Shenzhen Graduate School, Harbin Institute of Technology, Shenzhen 518055, China. [2] The Graduate Center, CUNY, New York, NY 10016, USA. [3] Department of Engineering Science and Physics, College of Staten Island, CUNY, Staten Island, NY 10314, USA. [4] Collaborative Innovation Center of Extreme Optics, Shanxi University, Taiyuan 030006, China. Correspondence and requests for materials should be addressed to L.G. (email: li.ge@csi.cuny.edu) or to S.X. (email: shumin.xiao@hit.edu.cn) or to Q.S. (email: qinghai.song@hit.edu.cn)

Lead halide perovskites (MAPbX$_3$, X = Cl, Br, I, or their mixtures) are an emerging class of semiconductors with a promising future in optoelectronic devices[1–4]. In past few years, driven by the continuous success in photovoltaics[3,5], other MAPbX$_3$ perovskite devices such as photodetectors[6–8] and light-emitting diodes[9–11] have also been rapidly developed. Perovskite microlasers are another prominent example and expected to have important applications in optical and quantum networks[12–36]. Soon after the discovery of exceptional gain in 2014[13,14], MAPbX$_3$ perovskite microlasers have been intensively studied in single-crystalline microstructures[15–19] and poly-crystalline films[20–22]. The threshold and quality (Q) factors of perovskite microlasers have been improved to 220 nJ cm$^{-2}$ [17] and $1 \times 10^4$ [12], respectively. With the progresses in nanofabrication techniques, top-down fabricated perovskite microlasers have also been experimentally realized in circular microdisks[12,22,24], distributed feedback structures[25–30], and even metasurfaces[35]. Very recently, continuous-wave (CW) perovskite microlasers have also been demonstrated at low temperature around 100 K[30], making room-temperature CW perovskite lasers very promising. Despite these exciting progresses, the demonstrated perovskites lasers are mostly static in use and their real applications are strongly hindered.

The footprints of micro- and nano-lasers are typically too small to implement additional control elements[12,36]. Thus fundamental mechanisms such as linear mode coupling and bistability were developed to control the lasing action[37–41]. In principle, these techniques are strongly dependent on the precise control of cavity sizes and resonant wavelengths, and thus are incompatible with lead halide perovskites because of their instability in polar solvents. The simple techniques including anion exchange[42–44], infiltration of liquid crystals[45], and phase transition of crystals at low temperature[30] can of course tune the lasing wavelengths. But they either are too slow for optical communications or require extreme conditions. To date, the mechanism for in situ and ultrafast control of perovskite microlasers is still absent and a breakthrough in fundamental mechanism is highly desirable. In this research, we explore nonlinear modal interactions in perovskite microlasers and demonstrate their impact on ultrafast mode switching, especially with cross-gain saturation[46,47] where the intensity of one lasing mode reduces the available gain for all other modes in the same system.

## Results

**Synthesis and characterization of MAPbBr$_3$ microrods.** Lead halide perovskite microrods were synthesized with a solution-based precipitation method[12] (see Methods). The top-view scanning electron microscope (SEM) image depicted in Fig. 1a shows that a large number of MAPbBr$_3$ perovskite microrods were synthesized simultaneously. The lengths, widths, and thicknesses are statistically analyzed (see Supplementary Fig. 1). They are around 10 μm, 800 nm, and 800 nm, respectively. The single crystal nature of synthesized perovskite microrods were determined by the following X-ray diffraction spectrum and high-resolution transmission electron microscopy investigation (see Fig. S2). Figure 1b shows the absorption and photoluminescence spectra of MAPbBr$_3$ perovskites microrods (see Methods, Supplementary Fig. 3 and Supplementary Fig. 4). A clear bandedge can be seen at ~2.32 eV, consistent with previous reports[12].

**The mode switching in MAPbBr$_3$ perovskite microlasers.** Then the laser characteristics of perovskite microrods were studied via optical excitation under a home-made microscope system (see Methods, Supplementary Fig. 3). When the microrod was entirely pumped, Fabry–Perot (FP) lasers along the axial direction were observed[17]. Once the microrod was partially excited, whispering gallery mode (WGM) lasers were usually formed in the transverse plane[19]. In general, lasing modes are in one-to-one correspondence with the passive cavity modes. Under a fixed excitation scheme (entirely or partially pumping), the main laser characteristics in a perovskite microrod are typically preserved very well during the lasing experiments.

Interestingly, there are also some novel perovskite microlasers that show different performances. Although the percentage of such lasers is quite low (detailed see Supplementary Note 4), they can still provide some hints for a new mechanism to control the perovskite microlasers. One example is depicted in Fig. 2. The tilt-view SEM image (inset in Fig. 2a) shows that the microrod has a rectangle cross-section with width and thickness of 1.67 and 1.81 μm. The length of microrod is 20 μm. The microrod was transferred to a clean substrate and placed onto another microrod with one end suspended in the air (see Fig. 2a) via micro-manipulation. In the optical experiment, only the suspended part was excited by adjusting the relative position between microrod and pumping laser spot (with $R \sim 20$ μm). At a low-pumping density, a broad photoluminescence peak appeared at 540 nm. With the increase of pumping density to 3.36 μJ cm$^{-2}$, one sharp peak (mode-1) emerged at 548 nm and rapidly dominated the emission spectrum at higher pumping density (see Fig. 2b). The corresponding full with at half maximum (FWHM) also reduced from 30 to 0.5 nm. The crosses in Fig. 2(c) depict the integrated intensity of mode-1 as a function of pumping density. When the pumping power was above 3.36 μJ cm$^{-2}$, a drastic increase in emission intensity was observed, indicating the onset of lasing actions in perovskite microrod. Due to the strong scattering loss and leakage at the overlapping region, the longitudinal Fabry–Perot modes in microrods will be suppressed[19]. Meanwhile, since the ends of two modes are widely separated, the coupling between transverse WGMs in two rods are also negligible. In this sense, the observed lasing actions shall be formed by the transverse WGMs in the top microrod only. This information can be simply confirmed with the fluorescent microscope image. As depicted in the inset of Fig. 2c, two bright laser spots can only be observed at the suspended end.

The interesting phenomenon occurred by further increasing the pumping power. When the pumping density was above 3.8 μJ cm$^{-2}$, the initial lasing peak gradually reduced and a new peak at 562 nm (mode-2) emerged (see Fig. 2b). To better understand the evolution between two lasing modes, we have finely tuned the pumping power in small steps and recorded the emission spectra. All the results are summarized in Fig. 2c. With the increase of pumping power, the intensities of peaks at 548 and 562 nm crossover. The latter one becomes the dominant peak above 4.31 μJ cm$^{-2}$ and the initial peak vanishes rapidly via a negative power slope. It is worth noting that the emission intensity of mode-2 shows a slight drop with further increase of pumping density, which is caused by Auger recombination at high pumping fluence. This limitation can be reduced with additional technique such as covering the sample with few-layer graphene[48]. Figure 2d shows the corresponding extinction ratio (defined as $10 \log(I_1/I_2)$) of the spectrum. The extinction ratio changed linearly from 15 to −15 dB with the increase of pumping density. The inset of Fig. 2d shows the polarization of these two WG modes, which are both transverse electrically (TE) polarized with dominant electric field perpendicular to the light propagation direction in the cross-sectional plane.

## Discussion

All of the above observations are intrinsically different from typical mode competition and clearly demonstrate the mode

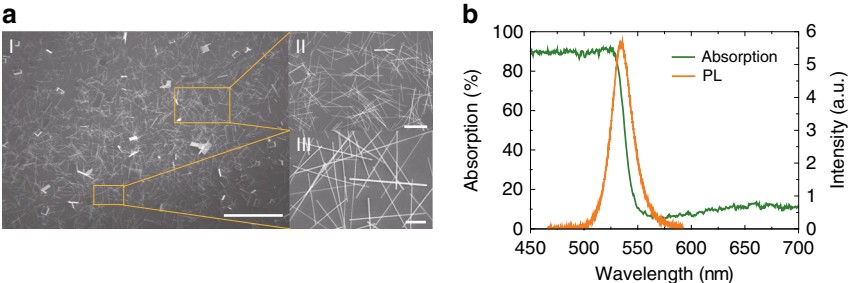

**Fig. 1** The synthesized MAPbBr$_3$ perovskite microrods. **a** The top-view SEM image of MAPbBr$_3$ microrods. The length of scale bar of I, II, and III are 100, 20, and 5 μm, respectively. **b** The absorption (green line) and photoluminescence (orange line) of MAPbBr$_3$ perovskite microrods

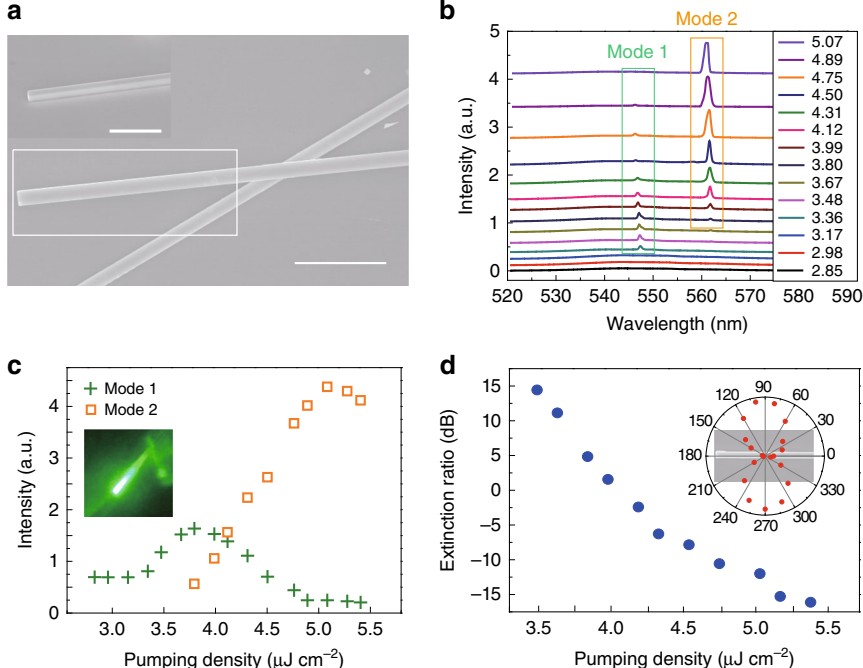

**Fig. 2** The optically switchable perovskite microlasers. **a** The SEM image of the perovskite microrod. Inset: tilt-view SEM image. The length of two scale bar are both 10 μm. **b** The emission spectra of the perovskite microrod at different pumping density (μJ cm$^{-2}$). **c** The integrated output of mode-1 (crosses) and mode-2 (squares) as a function of pumping density. Inset: Fluorescent microscope image of the microrod. **d** The extinction ratio of two lasing modes. Inset: The polarization of emitted light

switching in perovskite microlasers. Therefore, it is very interesting and important to explore the underlying mechanisms. In the literatures, there are several kinds of mechanisms that can generate mode switching. We can simply rule out the pump-dependent linear coupling between the two modes; it requires a noticeable change of the refractive index with the pump power or the change of its spatial profile[40], and hence would have significantly changed the frequency spacing between these two modes, for both strong coupling and weak coupling[49]. Instead, we have observed a constant frequency spacing (see Supplementary Fig. 8). And the 14 nm mode spacing is too large for typical linear mode coupling. Another possible mechanism is the optical bistability[37–39]. Figure 3a summarizes the lasing spectra as a function of pumping power. With the increase of pumping power from 3.36 to 5.07 μJ cm$^{-2}$ and then back to 3.17 μJ cm$^{-2}$ (see Fig. 3a), it is clear that the transition from mode-1 to mode-2 and back to mode-1 shows no hysteresis. More quantitatively, we plotted the extinction ratio of the spectrum along the loop in Fig. 3b, and the absence of bistability is clearly reflected by the good left–right symmetry. In addition, the mode switching behavior we observed is also different from mode hopping that manifests as a random

jump of the lasing peak between two modes. Mode hopping is driven by fluctuations such as the spontaneous emission noise and can take place at the same pump power[41]. In contrast, here the mode switching behavior was deterministic and did not occur when the pump power was fixed.

In our experiment, as the spatial pumping profile was fixed, we neglected the influence of evolving pumping profile. Below we turn to consider the influences of nonlinear modal interactions. In principle, the modal intensities $I_1$ and $I_2$ of two lasing modes can be understood with the following two-mode model[50] (see detail deviations in Supplementary Note 3)

$$M\begin{pmatrix} I_1 \\ I_2 \end{pmatrix} = \begin{pmatrix} \frac{D_0}{D_0^{(1)}} - 1 \\ \frac{D_0}{D_0^{(2)}} - 1 \end{pmatrix}, \quad M \equiv \begin{pmatrix} \Gamma_1 \chi_{11} & \Gamma_2 \chi_{12} \\ \Gamma_1 \chi_{21} & \Gamma_2 \chi_{22} \end{pmatrix} \quad (1)$$

Here, $I_\mu \geq 0$ ($\mu = 1, 2$), $\Gamma_\mu = \gamma_\perp^2 / \left[ \gamma_\perp^2 + \left( \omega_\mu - \omega_a \right)^2 \right] \leq 1$ is the Lorentzian gain factor for mode $\mu$ and $D_0^{(\mu)}$ is the threshold of mode $\mu$ without considering modal interaction. Mode-1 has a lower threshold than mode-2 ($D_0^{(1)} < D_0^{(2)}$), and $\omega_a$ is the atomic

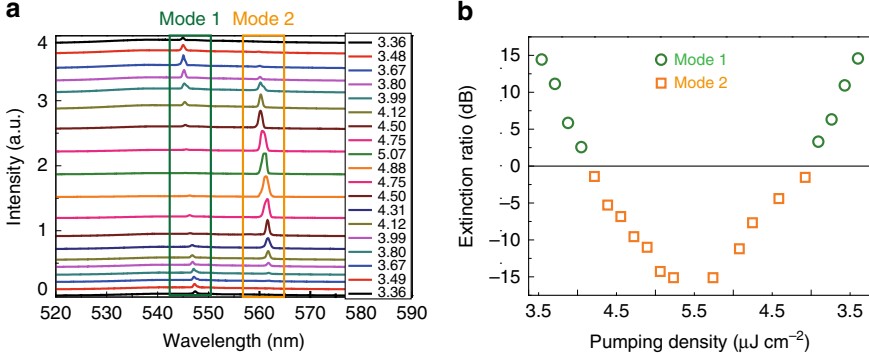

**Fig. 3** The reversible process. **a** The evolution of lasing spectrum with the increase and decrease of pumping density (μJ cm⁻²). **b** The corresponding extinction ratio in (**a**)

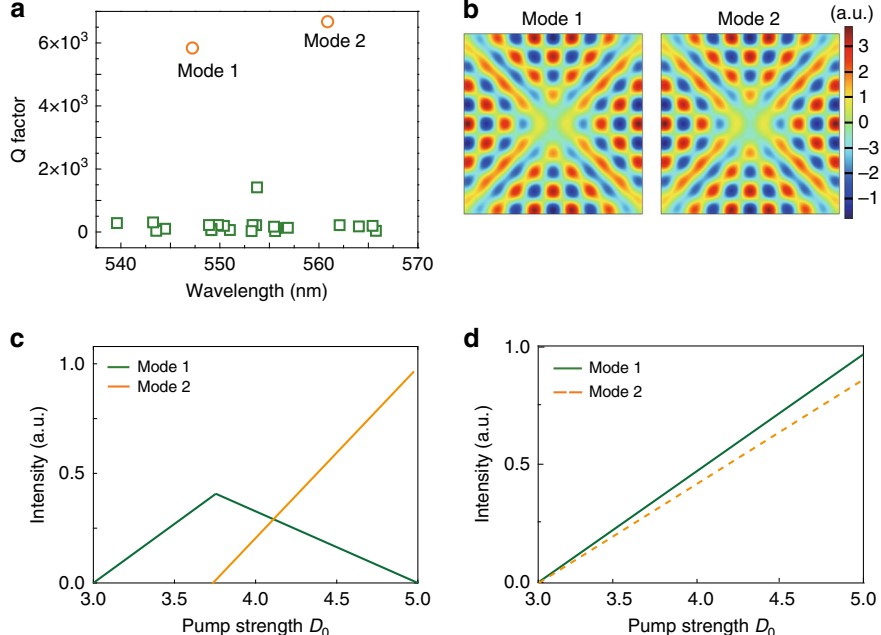

**Fig. 4** The mechanism for mode switching. **a** The numerically calculated Q factors in the cross-section of MAPbBr₃ perovskite microrod. **b** The field patterns of two high Q resonances. **c** The output intensity of two modes as a function of pumping power when the modal interaction is considered. **d** The control simulation without considering the modal interaction

transition frequency. $D_0$ is the pump strength and $\gamma_\perp$ is the longitude relaxation rate of the gain medium. $\chi_{11}$, $\chi_{22}$ are the self-interaction coefficients and $\chi_{12}$, $\chi_{21}$ are the cross-interaction coefficients, which are given by

$$\chi_{\mu\nu} = \frac{1}{V}\left| \int_{\text{cavity}} d\mathbf{r} u_\mu^2(\mathbf{r})|u_\nu(\mathbf{r})|^2 \right| \qquad (2)$$

where $V = \int_{\text{cavity}} d\mathbf{r}$ and $u_\mu(\mathbf{r})$ is the normalized and dimensionless field distribution of mode-μ in the cavity. Then the parameters in the above equation were studied with numerical calculations and SALT theory (see Methods and Supplementary Note 3). In our calculations, the structural parameters followed the SEM image, whereas the refractive index (n) and light extinction coefficient (k) were measured by ellipsometer experimentally (see Supplementary Fig. 9). Figure 4a summarizes the TE (see inset in Fig. 2d) polarized resonances within the transverse plane, where tens of resonances can be seen. Most of the resonances are FP modes with low-Q factors except two modes with similar and relatively high-Q factors. The corresponding

field patterns in Fig. 4b show that these two modes are both 4-bounce WGMs. Following the SALT calculation, the non-interaction thresholds of two modes were $D_0^{(1)} = 3\,\mu\text{J cm}^{-2}$ and $D_0^{(2)} = 3.0004\,\mu\text{J cm}^{-2}$. The self-interaction coefficients $\chi_{11}$, $\chi_{22}$ and the cross-interaction coefficients $\chi_{21}$, $\chi_{12}$ were calculated from Eq. (2). They are $\chi_{11} = 3.8648 \times 10^{-6}$, $\chi_{12} = 3.862 \times 10^{-6}$, $\chi_{21} = 3.862 \times 10^{-6}$, and $\chi_{22} = 3.8607 \times 10^{-6}$.

According to Eq. (1), the power slope of the first mode is $S_1 = 1/\left|\Gamma_1\chi_{11}D_0^{(1)}\right|$ at $D > D_0^{(1)}$. Once the second mode starts to lase beyond its own threshold, the slope of the initial mode changes to

$$\widetilde{S}_1 = \frac{\frac{\chi_{11}}{\chi_{21}} - \frac{D_0^{(2)}}{D_0^{(1)}}}{\frac{\chi_{11}}{\chi_{21}} - \frac{\chi_{12}}{\chi_{22}}} S_1. \qquad (3)$$

By using the above values for $\chi_{\mu\nu}$, it is easy to see that the power slope $\widetilde{S}_1$ becomes negative in Eq. (3). Figure 4c illustrates the modal intensities of two lasing modes as a function of $D_0$. We find that mode-1 lases first and its modal intensity increases until

mode-2 reaches the threshold. Further increasing $D_0$ increases the intensity of mode-2 quickly, while the power slope of the first lasing mode changes to a negative value. All of these behaviors are consistent with our experimental results and show that the modal interaction played a crucial role in the mode switching. For a direct comparison, we have also calculated the power slopes without considering the cross-interaction. As shown in Fig. 4d, these two modes, which have almost identical noninteracting thresholds $D_0^{(1)}$, $D_0^{(2)}$, turn on almost simultaneously and both maintain a positive power slope as the pump power increases. This is intrinsically different from our experimental observations.

Following the Eq. (3), the criterion for modal interaction induced mode switching can be derived as

$$\frac{\chi_{21}}{\chi_{11}} < \frac{\chi_{22}}{\chi_{12}} < \frac{D_0^{(1)}}{D_0^{(2)}}. \qquad (4)$$

This criterion is quite generic and provides a simple rule to select the perovskite nanorod for mode switching. Basically, two requirements must be fulfilled. The noninteraction thresholds of two modes, which are determined by cavity Q factors and the gain spectrum, must be very close without considering the modal interaction. Meanwhile, the mode profiles of two modes shall largely overlap to fulfill Eq. (4). The above criterion can guide us to search and realize mode switching in perovskite nanorods.

One example is illustrated in Fig. 5. As the side-view SEM image depicted as inset in Fig. 5c, the new perovskite microrod is largely rectangular with surface roughness increasing from left to right. The corresponding photoluminescence experiment shows that the right part was not well grown and converted to MAPbBr$_3$ perovskites. As a result, the right end purely provides strong scattering loss and only random lasers along axial direction can be formed. Following the SEM image, we have numerically studied the resonances within the nanorods. Air holes were introduced to represent the surface roughness shown in the SEM. Two modes have been obtained with similar Q factors higher than the others. The insets in Fig. 5b show their mode profiles. We can see that two modes that are mainly confined in the left half, ensuring their spatial overlap and the corresponding modal interaction induced mode switching. Figure 5b shows the calculated intensities of two resonances following SALT theory. A clear mode switching process can also be observed.

Then the laser characteristics of the microrod was also studied with optical excitation. The perovskite microrod exhibited a broad photoluminescence peak at low-pump power. When the pump power was 4.8 μJ cm$^{-2}$, a lasing peak at 546.5 nm (mode-1) could be clearly observed (black line in Fig. 5a). Once the pump power was further increased to 5.2 μJ cm$^{-2}$, the initial lasing peak was suppressed and a new lasing peak at 556 nm (mode-2) appeared (red dashed line in Fig. 5a). Figure 5c summarizes the integrated output intensity as a function of pump power. Mode-1 lased first and dominated the laser spectrum at the beginning. Once mode-2 turned on, mode-1 experienced a negative power slope and almost disappeared at around 5.4 μJ cm$^{-2}$. During this process, the extinction ratio between two modes changed from 10 to −10 dB (see Fig. 5d). The corresponding fluorescent microscope image (inset in Fig. 5d) also shows that the lasers were well trapped within the left parts. All of these observations are consistent with the numerical calculations in Fig. 5b and confirm the above criterion well. In experiments, the preferable condition that leads to this phenomenon cannot be satisfied by all types of modes, and based on this strict requirement of the criterion and the random sizes of as-grown, mode switching is not ubiquitous in the as-grown perovskite microlasers (detailed see

Supplementary Note 4). This can be solved by high-quality top-down nanofabrication technique to precisely fabricate the designed cavities (see Supplementary Note 8).

In additional to the switch in wavelengths, it is also interesting to study the response in the time domain, which is a key parameter for applications in optical communications and quantum information. In our experiment, we have studied the switching time by two pump pulses with variable delay time (τ). The experimental setup is shown in the Supplementary Fig. 10. As depicted in Fig. 6a, the pumping powers of two laser pulses are designed in the following way. One pulse (pulse-1) can excite mode-1, whereas the other one (pulse-2) can only generate the spontaneous emission. However, the superposition of two pulses is large enough to switch the mode-1 to mode-2. The inset of Fig. 6a is the SEM image of a new perovskite sample, where the length of scale bar is 10 μm. The lasing actions were also generated in its transverse plane. The mode intensity as a function of time delay is summarized in Fig. 6b. With the decrease of delay time, the intensity of mode-1 increased first and decreased rapidly at around τ = −125 ps. The microlaser was dominated by mode-1 at τ < −125 ps and only the mode-2 was observed at τ > −50 ps. The extinction ratio was plotted in Fig. 6c. We consider the change of 10 dB (5 to −5dB) as the main switching process. According to the results in Fig. 6c, the switching time is around 75 ps, which is fast enough for a lot of practical applications.

In summary, we have studied the lasing actions in lead halide perovskite microrods. In contrast to the previous reports of static microlasers, we show that a single-mode laser can be switched off by varying the excitation density, making the perovskite microlasers dynamically switchable for the first time. The experimental observations and the corresponding theoretical analysis reveal that strong modal interactions play a key role in this mode switching process. In addition, the speed of interacting-induced mode switching is found to be less than 75 ps. Our results have provided a simple and robust approach for all-optically controllable microlasers. This mechanism is not limited in MAPbBr$_3$ perovskites. It also works well in other materials such as polymer. Importantly, with the mature top-down fabrication technique of polymer, this mode switching phenomenon can be well reproduced in a series of polymer microcavities (see Supplementary Note 7), indicating that the low production rate of switchable perovskite lasers can be eventually improved with the development of top-down fabrication techniques. This kind of mode switching may find practical applications in optical memory, flip-flop, and other functional devices such as Raman lasers and sensors.

## Methods

**Synthesis of perovskite microwires.** The lead halide perovskites were synthesized on hydrophobic indium tin oxides (ITO) coated glass with one-step solution self-assembly method. Basically, CH$_3$NH$_3$Br and PbBr$_2$ were independently solved in N, N-dimethylformamide (DMF) with concentrations around 0.1 M. Then two solutions were mixed at room temperature with 1.05:1 volume ratio to form CH$_3$NH$_3$Br·PbBr$_2$ solution (0.02 M). The diluted solution was dip-casted onto a ITO glass side, which was placed on a Teflon stage in 50 ml beaker. Totally, 35 ml dichloromethane (DCM) of CH$_2$Cl$_2$ was placed in the beaker and sealed with a porous Parafilm (3 M) to control the evaporation speed. After 48 h, lead halide perovskites (CH$_3$NH$_3$PbBr$_3$) have been successfully synthesized on the substrate.

**Lasing measurement.** The experimental setup of optically pumped lasing measurement is shown in Supplementary Fig. 2. The perovskite samples were mounted onto a three-dimensional translation stage under a home-made microscope and excited by a frequency doubled laser (400 nm, using a BBO crystal) from a regenerative amplifier (repetition rate 1 kHz, pulse width 100 fs, seeded by MaiTai, Spectra Physics). The pump light was focused onto the top surface of the samples through a 40× objective lens and the radius of the beam size was adjusted to R ~20 μm. The emitted lights were collected by the same objective lens and coupled to a CCD (Princeton Instruments, PIXIS UV enhanced CCD) coupled spectrometer (Acton SpectroPro s2700) via a multimode fiber. All the emission spectra were

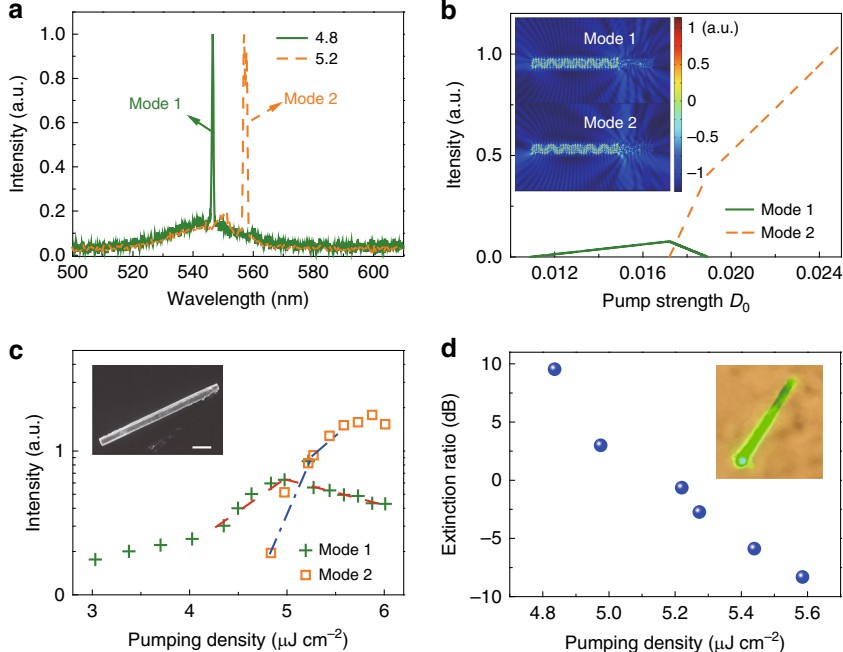

**Fig. 5** Mode switching in a single perovskite microrod. **a** Lasing spectra at different pump intensities. **b** Numerical simulations of the mode switching. The parameters used here are: $k_aL = 114.3$ and refractive index $n = 2.56 - 0.001i$. The calculated self-interaction and cross-interaction coefficients are $\chi_{11} = 1.2582 \times 10^{23}$, $\chi_{12} = 3.0256 \times 10^{22}$, $\chi_{21} = 3.0256 \times 10^{22}$, and $\chi_{22} = 1.0403 \times 10^{22}$. The insets are the field patterns of two high-Q resonant modes. **c** Output intensities of the two modes as a function of the pumping density. Inset: SEM image of the perovskite microrod. The length of scale bar is 5 μm. **d** Extinction ratio as a function of the pumping density. Inset: Fluorescent microscope image of the microrod

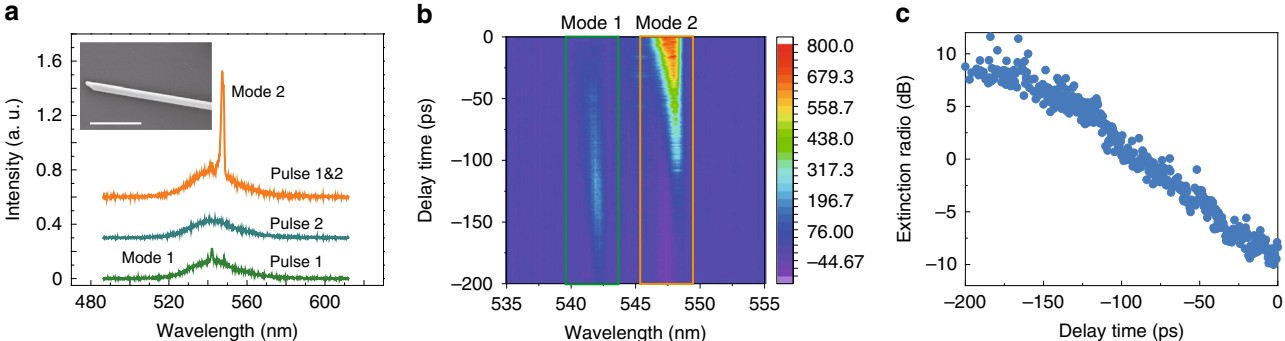

**Fig. 6** Temporal response of mode switching in perovskite microrod. **a** Output intensity of perovskite microrod by different pulses. Inset: the SEM image, where the length of scale bar is 10 μm. **b** Mode intensity as a function of delay time. **c** Relative intensity of the two modes (mode-1/mode-2) as a function of delay time

measured using a 150 g mm$^{-1}$ grating with 0.3-nm resolution. The fluorescent microscope images were recorded by a CCD camera behind a longpass filter.

**Numerical simulation**. The numerical calculations were performed with commercial finite-element-methods based software (Comsol Multiphysics 5.3a). The cross-section and the axial direction of the microrods were treated as two-dimensional objects with an effective refractive index. The openness of the system was simulated with a perfectly matched layer to absorb the outgoing waves without reflection. Thus the calculated frequencies of quasi-bound states were complex numbers. The calculated frequencies and field patterns of modes were used as inputs in SALT[49–54] to study their lasing actions. Similar to the experiments, only TE (E in plane) field was considered in the numerical calculations.

## Data availability

The data that support the findings of this study are available from the corresponding author upon reasonable request.

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

## Acknowledgements

This research is financially supported by National Natural Science Foundation of China under the grant No. 91850204 and Shenzhen Fundamental research projects under the grant No. JCYJ20160427183259083.

## Author contributions

Q.S., S.X., and L.G. conceived the idea and designed the research. N.Z., Y.F., K.W., Z.G. and Y.W. fabricated the samples and optically characterized the lasing actions. N.Z., Y.F. and K.W. measuring the temporal response of lasing actions. N.Z., Z.G. and L.G. do the numerical simulation. N.Z., Y.F., K.W., L.G., S.X. and Q.S. discussed the results and wrote the paper. N.Z., Y.F. and K.W. contributed equally to this work.

## Additional information

**Competing interests:** The authors declare no competing interests.

