## [Peer Review File · Nature Communications]

Reviewers' comments:

Reviewer #1 (Remarks to the Author):

This manuscript reports a mechanism to optically switch the operation wavelength of MAPbBr₃ perovskite nanorod lasers. The strategy is based on two coupled lasing modes tuned with the excitation fluency. The experimental results are adequately corroborated by the theory and simulations, and at the end the author propose a time domain response of the switching. The paper is well organized, figures are clear and conclusions are well extracted. Indeed, I found this is an interesting approach that could be of interest to the audience of Nature Communications. The manuscript, however, present also some drawbacks that prevent the publication in Nature Communications in its current form.

1. First of all, both the experiment and final device are not completely clear for me. According to the sentence " The microrod was placed onto another microrod with one end suspended in air via the micro-manipulation " I guess that there are two microrods coupled by the end edges, but then only one microrod is described or showed in the figures. Authors should include a scheme of the experiment indicating clearly where the excitation and the collection are. Here it is also necessary to specify the size of the spot in excitation and collection. Are both the same, 20 μm? Are authors studying just one rod or an ensemble? What is the concentration of rods per cm²

2. The second paragraph of section 2 is a bit confusing. First, authors argue that they choose FP or WGM by exciting entirely or partially the the microrod, but then they say that "In general, both modes are observed" What does it mean? Are Fabry Perot (FP) modes or Whispering Gallery Modes (WGM)? How do they observe the modes experimentally? Can authors show an experimental near field pattern of the modes?

After reading the description of Figure 4 I understood that they are working with 2 WGM modes with practically the same pattern. Then, are they working with only one microrod instead of a coupled cavity system? If it is necessary to change the excitation area to observe the WGM modes, I understand that spot size is changed. Does this change affect the measurements since excitation fluency would be different? Can authors specify which modes present the highest Q factor (TE_x)?

3. I also do not understand the sentence "without carefully designing the fundamental mechanism such as bistability" I think here authors want to claim that they obtain a stable laser device through a straightforward fabrication or without complicated design of the cavity. However, in this section of the paper the switching results are not yet introduced, hence the sentence could be confusing for the reader.

4. The sentence "Interestingly, there are also some novel perovskite microlasers that can provide some hints to control the perovskite microlasers" needs to be rewritten. Do authors mean that some microrods work and others does not work? They should indicate some percentage in that case.

5. The switching is just based on an increase of the excitation fluency. Since this is in principle a simple activation mechanism, I understand that it should not be difficult to observe this behavior once the threshold of stimulated emission is overcome. Nevertheless, there are several publications where lasing in MaPbX₃ nanorods/microdisk is reported from authors' team (References 20,22 or others) or others. Although these works use similar excitation conditions (femtosecond pulses), apparently this is the first time where the optical switch is observed. Why? Is there any difference between the proposed experiments compared with those developed in the previous works? Here, authors should remark what is new in their approach and why this approach let them to observe this switch. In my opinion, this is very important for a Nature Communications paper. In addition, I miss a comparison of these results (threshold, gain...) with previous publications. For this purpose, I think is better if excitation power is specified as excitation fluency (power per area). Indeed, if the diameter of the spot is 20 μm the excitation fluency at the threshold becomes $0.53/\pi/100=170$ mJ/cm², which is rather high. Are some numbers wrong here?

5. Is this strategy developed previously with other materials?

I have also found other minor scientific/style comments that should also be addressed:

6. I miss in the supporting information the spectra below and above the lasing threshold. Here I

think it should be better to show the Y-axis in a logarithmic scale. In addition, it would be interesting to include in the supporting information the integrated intensity as a function of the excitation fluency in a log-log plot.

7. Absorption curve in Figure 1 indicates strong scattering at long wavelengths. Are results affected by this scattering? What is the size of the spot in the absorption measurements?
8. How n and k are measured?
9. I do not understand the section VII included in the supporting information. This section is even longer than the main manuscript and it is only called in the conclusions.
10. I found misprints in line 70 "confim" by "confirms", 109 mechanism by mechanisms, line 169 "sepctrum" by spectrum.
11. Reference 39 is missing.
12. Line 56. Include a reference about cross gain saturation observed with other materials, and explain what it is.
13. I do not understand why the inset of Figure 2d is referred in Line 142. Anyway, this inset is not commented within the Figure (lines 95-104).
14. Line 149. Are cross-section interaction coefficients normalized magnitudes? or are units missed?

Reviewer #2 (Remarks to the Author):

Comments to the Authors

In the manuscript entitled "All-optical control of lead halide perovskite microlasers", N. Zhang, et al. reported 70-ps ultrafast all-optical mode switching based on organic-inorganic lead halide perovskite (MAPbBr₃) microlasers. By adjusting the excitation energy, mode switching with high extinction ratio (~15 dB) between two distinct modes has been demonstrated. The experimental observations have been well explained with a cross-gain saturation model, which is also applicable to other microlasers (e.g. polymer microdisk lasers in the supplemental information). All-optical control on perovskite microlasers is currently the subject of extensive research efforts by many groups. The mechanism in this manuscript is timely and important for the developments of perovskite-based photonics in near future. All the experimental observations and theoretical discussion are interesting, convincing, well organized and accessible to a broad audience. Therefore, I recommend the acceptance of the manuscript after some minor revisions.

- 1) The references in the introduction section should be carefully checked, e.g. though the introduction mainly focuses on MAPbX₃, some works on all inorganic perovskites have been cited; on Line-41, Page-2, the references 28-33 are not about photonic crystals; the reference 39 is missing in the reference list.
- 2) In Line-38, Page-2, the record values on threshold and Q factor of perovskite microlasers cannot be found in the reference. The authors should very accurate information or citation.
- 3) In Figure 1, authors mainly discuss the crystalline quality and optical properties of perovskites, which are not directly relevant to the topic of the manuscript. Thus I suggest that authors add them into the supporting information.
- 4) The description on the excitation is not proper: pumping density should be changed to pumping energy, considering that joules are used in the manuscript to describe the excitation.
- 5) In Figure 2c, the intensity of mode 2 slightly drops when the excitation energy is above 0.8 μ J. What is the cause of the drop? It seems that the Auger effect starts to play an important role. The author should discuss this and give a possible way to solve this effect.
- 6) Do the two competing modes in Figure 2b have the same polarization? The polarization shown in the inset of Figure 2d has not been described with detail information.
- 7) Typos and grammar errors should be checked through the whole manuscript, e.g. on Line-21, Page-1, "reversable" should be "reversible"; on Line-22&23, Page-1, there are two redundant hyphens.

Reply to Reviewer #1:

We thank the reviewer for the careful review and valuable suggestions to improve the quality of our research. According to the reviewer's comments, the manuscript has been carefully revised and its quality has been improved greatly.

Comment-1 First of all, both the experiment and final device are not completely clear for me. According to the sentence "The microrod was placed onto another microrod with one end suspended in air via the micro-manipulation" I guess that there are two microrods coupled by the end edges, but then only one microrod is described or showed in the figures. Authors should include a scheme of the experiment indicating clearly where the excitation and the collection are. Here it is also necessary to specify the size of the spot in excitation and collection. Are both the same, 20 μm ? Are authors studying just one rod or an ensemble? What is the concentration of rods per cm^2 .

Our Reply: We thank the reviewer for this careful review and valuable suggestions. We are sorry for these confusions. In our experiments, all the experimental phenomena were obtained from one microrod every time. No coupling effect has been taken into account.

The two-microrod scheme was initially selected following Ref. 19. One rod was placed onto the other one and had one end suspended in air (see Fig. R1). As the refractive indices of two perovskite microrods are exactly the same, the guiding modes in the longitude direction shall experience significant scattering loss and leakage at the joint position. Then the longitude Fabry-Perot modes are completely suppressed and only the transverse WGMs can have high Q factors. In our experiment, only the suspended end of the top microrod was excited to generate laser emission (see Fig. R2). This has been confirmed by the fluorescent microscope image in the inset of Fig. 2(c) in the main manuscript, where bright spots are only observed at the suspended end of the top microrod. According to the SEM image, the ends of two microrods are separated tens of wavelengths, the coupling between transverse modes in two microrods can thus be neglected.

Figure R1: The SEM image of the two microrods system. The dashed

With the progress of our experiment, we also realized that the material absorption of perovskite at the band edge was very large when the microrod was not optically pumped. In this sense, if only one end of a perovskite microrod was pumped, the longitude Fabry-Perot modes could also be suppressed by the material absorption at un-pumped area. In some experiments, we only utilized a single microrod with partial pumping (see schematic picture

in Fig. R2) and obtained the switching effect, e.g. the experimental results such as Fig. 6 of the main manuscript.

To avoid the possible confusion, we have replaced the SEM image in Fig. 2(a) with Fig. R1. The original tilt-view SEM is added as an inset in Fig. 2(a). We have also added the corresponding description in para-3, page-3. “The microrod was transferred to a clean substrate and placed onto another microrod with one end suspended in the air (see Fig. 2(a)) via micromanipulation. In the optical experiment, only the suspended part was excited by adjusting the relative position between microrod and pumping laser spot (with \$R \sim 20 \mu\text{m}\$ ). At a low pumping density, a broad photoluminescence peak appeared at \$540 \text{ nm}\$.” and “Due to the strong scattering loss and leakage at the overlapping region, the longitude Fabry-Perot modes in microrods will be suppressed¹⁹. Meanwhile, since the ends of two modes are widely separated, the coupling between transverse WGMs in two rods are also negligible. In this sense, the observed lasing actions shall be formed by the transverse WGMs in the top microrod only. This information can be simply confirmed with the fluorescent microscope image. As depicted in the inset of Fig. 2(c), two bright laser spots can only be observed at the suspended end.”

Figure R2: Schematic diagram of the lasing experiment. (This figure has been added as inset of Fig. S3).

Following the reviewer’s suggestions, we have also added the experimental details in the main manuscript. The excitation beam was focused on the perovskite microrod through a 40× objective (NA=0.6) and spot radius is around $20 \mu\text{m}$. The emitted light was collected through the same objective and the collection area was much larger than the pumped region. The corresponding descriptions on the excitation and collection scheme have been included in the Methods section. “The perovskite samples were mounted onto a three-dimensional translation stage under a home-made microscope and excited by a frequency doubled laser (\$400 \text{ nm}\$, using a BBO crystal) from a regenerative amplifier (repetition rate \$1 \text{ kHz}\$, pulse width \$100 \text{ fs}\$, seeded by MaiTai, Spectra Physics). The pump light was focused onto the top surface of the samples through a 40x objective lens and the radius of the beam size was adjusted to \$R \sim 20 \text{ micron}\$. The emitted lights were collected by the same objective lens and coupled to a CCD (Princeton Instruments, PIXIS UV enhanced CCD) coupled spectrometer (Acton SpectroPro 2700) via a multimode fiber. All the emission spectra were measured using a \$150 \text{ g/mm}\$

grating with 0.3-nm resolution. The fluorescent microscope images were recorded by a CCD camera behind a longpass filter.”

During the experiment, only one microrod was studied. From the SEM image below (Figure R3a), hundreds of perovskite microrods can be seen in a small area about $100\ \mu\text{m} \times 100\ \mu\text{m}$. To eliminate the influence of other perovskite microrods, only one perovskite microrod was selected by an optical fiber tip and transferred to a clean wafer by micro-manipulation (Figure R3b, Wang, K. et al. Formation of single-mode laser in transverse plane of perovskite microwire via micromanipulation. Opt. Lett. 41, 555-558 (2016)). This information has also been added in para-3, page-3. “The microrod was transferred to a clean substrate and placed onto another microrod with one end suspended in the air (see Fig. 2(a)) via micromanipulation.”

[Redacted]

[Redacted]

Figure R3: (Wang, K. et al. Formation of single-mode laser in transverse

plane of perovskite microwire via micromanipulation. Opt. Lett. 41, 555-558 (2016))

Comment-2 The second paragraph of section 2 is a bit confusing. First, authors argue that they choose FP or WGM by exciting entirely or partially the microrod, but then they say that "In general, both modes are observed" What does it mean? Are Fabry Perot (FP) modes or Whispering Gallery Modes (WGM)? How do they observe the modes experimentally? Can authors show an experimental near field pattern of the modes? After reading the description of Figure 4 I understood that they are working with 2 WGM modes with practically the same pattern. Then, are they working with only one microrod instead of a coupled cavity system? If it is necessary to change the excitation area to observe the WGM modes, I understand that spot size is changed. Does this change affect the measurements since excitation fluency would be different? Can authors specify which modes present the highest Q factor (TEx)?

Our Reply: We thank the reviewer for this careful review and valuable comments. We realized that we hadn't described the typical lasing actions in perovskite microrods clearly. In a microrod, two kinds of resonant modes with relatively high Q factors could be supported: Fabry-Perot (FP) modes along the axial direction of microrod; and WGMs formed by

four-bounce total internal reflection in the transverse plane of the microrod (MAPbBr₃ perovskite microrods have rectangular or even square cross-sections). Typically, the FP lasers were generated by entirely pumping the microrod (see Zhu et al. Nat. Mater. 14, 636-642 (2015)). As mentioned above, once the microrod was partially excited, the WGM laser can be generated. Both kinds of lasing modes are in one-to-one correspondence with the passive cavity resonances.

In our case, only the WG mode lasing were observed by partially exciting one end of the perovskite microrod as shown in Figure R2. To eliminate the misunderstanding, we have revised the writing in para-2, page-3 and added the references 17 and 19 for examples. “When the microrod was entirely pumped, Fabry-Perot (FP) lasers along the axial direction were observed¹⁷. In case that only one end of microrod was excited, whispering gallery mode (WGM) lasers were usually formed in the transverse plane¹⁹. In general, lasing modes are in one-to-one correspondence with the passive cavity modes. Under a fixed excitation scheme (entirely or partially pumping), the main laser characteristics in a perovskite microrod are typically preserved very well during the lasing experiments.”

Fabry-Perot lasers and WGM lasers from a perovskite microrod can be easily distinguished by the lasing spectrum or fluorescence microscope image. Figure R4 illustrates the experimental results of FP mode lasing (a) and WG mode lasing (b) by pumping the same perovskite microrod entirely and partially on one end (Wang, K. et al. Formation of single-mode laser in transverse plane of perovskite microwire via micromanipulation. Opt. Lett. 41, 555-558 (2016)). Since the resonant cavity length of FP mode is much larger than that of WG mode, the mode spacing of FP mode is thus much smaller than that of WG mode, according to $\Delta\lambda = \lambda^2/nL$ (λ is the resonance wavelength, n is the effective refractive index and L is the effective cavity length). Therefore, for the same perovskite microrod (length: 53.18 μm , width: 1.51 μm , thickness: 1.22 μm), FP mode lasing shows periodic multimode lasing peaks, whereas WG mode lasing shows a single lasing peak. Besides, FP modes are bouncing between two end-facets and thus show two bright spots at two ends of microrod (see inset in Fig. R4(a)). In contrary, the WGMs are confined with total internal reflection and shall only emit along the tangential line of each side. Thus we should observe the bright laser emission spots at the side-facets of the pumping area (see right inset in Fig. R4(b)). Based on the above information and the experimental observation in the inset of Fig. 2(c), we confirm that we observed the WGM lasers. We have added the corresponding discussion in para-1, page-4. “Due to the strong scattering loss and leakage at the overlapping region, the longitude Fabry-Perot modes in microrods will be suppressed¹⁹. Meanwhile, since the ends of two modes are widely separated, the coupling between transverse WGMs in two rods are also negligible. In this sense, the observed lasing actions shall be formed by the transverse WGMs in the top microrod only. This information can be simply confirmed with the fluorescent microscope image. As depicted in the inset of Fig. 2(c), two bright laser spots can only be observed at the suspended end.”

[Redacted]

[Redacted]

Figure R4: (Wang, K. et al. Formation of single-mode laser in transverse plane of perovskite microwire via micromanipulation. *Opt. Lett.* 41, 555-558 (2016)).

As mentioned in the above, the two competing modes are generated from the same perovskite microrod. The other microrod was utilized to completely suppress the FP mode in the microrod under investigation. And no coupling between different perovskite microrods has been involved. To eliminate the misunderstanding, the description on the pumping scheme has been revised in para-3, page-3. “The microrod was transferred to a clean substrate and placed onto another microrod with one end suspended in the air (see Fig. 2(a)) via micromanipulation. In the optical experiment, only the suspended part was excited by adjusting the relative position between microrod and pumping laser spot (with \$R \sim 20 \mu\text{m}\$ ). At a low pumping density, a broad photoluminescence peak appeared at 540 nm.”

In our case, the excitation spot (pumping position and spot size) on the perovskite microrod was fixed throughout the experiment. Thus the influence of variation of pumping size is also negligible. Furthermore, owing to the single crystalline nature of the microrod, we have previously proved that the transverse WGM lasers from different positions of the same microrod are identical (see Wang K et al *J. Phys. Chem. Lett.* 7, 2549-2555 (2016)). In this sense, even the pumping spot slightly changes a little bit, it won't affect the lasing actions in transverse plane of the single-crystalline microrod.

In the transverse plane, there are also numerous resonant modes in the transverse plane. Most of them are Fabry-Perot modes confined by reflection at the side facets. Thus the Q factors are very low. Only two of them are confined by total internal reflection and have higher Q factors. All the results are shown in Fig. 4 in the main manuscript. As all the structural parameters and refractive index in simulation are measured from experiments, we can see these two modes match the experimental results also well. The corresponding fluorescent microscope image in the inset of Fig. 2(c) is also consistent with the WGMs.

Comment-3 I also do not understand the sentence "without carefully designing the fundamental mechanism such as bistability" I think here authors want to claim that they obtain a stable laser device through a straightforward fabrication or without complicated design of the cavity. However, in this section of the paper the switching results are not yet

introduced, hence the sentence could be confusing for the reader.

Our Reply: We thank the reviewer for this careful review and valuable suggestion. We agree with the reviewer that the statement is quite confusing. Following the reviewer's suggestion, we have changed the sentence as follows (see para-2, page-3). "In general, lasing modes are in one-to-one correspondence with the passive cavity modes. Under a fixed excitation scheme (entirely or partially pumping), main laser characteristics in a perovskite microrod were typically preserved very well during the lasing experiments."

Comment-4 The sentence "Interestingly, there are also some novel perovskite microlasers that can provide some hints to control the perovskite microlasers" needs to be rewritten. Do authors mean that some microrods work and others does not work? They should indicate some percentage in that case.

Our Reply: We thank the reviewer for this valuable suggestion. The reviewer is absolutely right that we want to mention that some microrods can show quite different laser characteristics from the conventional ones. In experiments, this kind of nanorods are quite rare. We usually need to measure hundreds of microrods to get several samples. Thus the percentage is less than 1 percent.

Such a low production rate is understandable in as-grown samples. Basically, three conditions should be fulfilled to realize the mode switching. I) There must be two lasing modes. II) The non-interaction laser thresholds, which is typically determined by the Q factors, must be very close. III) The profiles of two modes must largely overlap to fulfill the equation, $\frac{\chi_{21}}{\chi_{11}} < \frac{\chi_{22}}{\chi_{12}} < \frac{D_0^{(1)}}{D_0^{(2)}}$, which is the criterion for modal interaction induced mode switching. According to our previous experimental results, the transverse sizes of microrods are usually so small that only single-mode WGM lasers have been obtained (see Wang, K. et al. Opt. Lett. 41, 555-558 (2016)). The other transverse modes have much lower Q factors. In addition, the criterion $\frac{\chi_{21}}{\chi_{11}} < \frac{\chi_{22}}{\chi_{12}} < \frac{D_0^{(1)}}{D_0^{(2)}}$ requires that two modes largely overlap one another to ensure strong cross-interactions. This is usually not easy to be fulfilled. We can simply illustrate this strict requirement with the microrod in Fig. 4 of the manuscript. As shown in Fig. 5(a), an additional high Q mode at 533.26 nm can also exist in the transverse plane of microrod. While its mode profile (inset in Fig. R5) is still very close the ones in Fig. 4(b), the interaction between mode-1 and mode-3 cannot fulfill the criterion. In other world, if mode-1 and mode-3 are excited in experiment, they will show a regular mode competition instead of mode switching (see Fig. R5(b)) although mode switching has been observed between mode-1 and mode-2 in the same microrod. Therefore, based on the strict requirement of the criterion and the random sizes of as-grown, mode switching is not ubiquitous in the our as-grown perovskite microlasers.

Figure R5: (a) The numerically calculated Q factors in the cross-section of MAPbBr₃ perovskite microrod. (b) The output intensity of mode 1 and 3 as a function of pumping power when the modal interaction is considered.

It is worth to note that the mechanism for mode switching is robust and can be repeated. This requires a high-quality top-down nanofabrication technique to precisely fabricate the designed cavities. The examples are shown in Supplementary Figures 8, 9, 16 and 17. With the top-down fabrication techniques, the mode switching has been repeated in a series of polymer microlasers. Therefore, while the mode switching is quite rare in the as grown samples, it can be generic in perovskite microlasers if the top-down fabrication technique of perovskite is further improved.

Following the reviewer's suggestion, we have rewritten the first sentence of para-3, page-3 in the revised manuscript. "Interestingly, there are also some novel perovskite microlasers that show different performances. Although the percentage of such lasers is quite low, they can still provide some hints for a new mechanism to control the perovskite microlasers." We have also added the comment of production rate in para-1, page-9. "This mechanism is not limited in MAPbBr₃ perovskites. It also works well in other materials such as polymer. Importantly, with the mature top-down fabrication technique of polymer, this mode switching phenomenon can be well reproduced in a series of polymer microcavities (see Section VII in Supplementary), indicating the low production rate of switchable perovskite laser can be eventually improved with the development of top-down fabrication techniques."

Comment-5 The switching is just based on an increase of the excitation fluency. Since this is in principle a simple activation mechanism, I understand that it should not be difficult to observe this behavior once the threshold of stimulated emission is overcome. Nevertheless, there are several publications where lasing in MAPbX₃ nanorods/microdisk is reported from authors' team (References 20, 22 or others) or others. Although these works use similar excitation conditions (femtosecond pulses), apparently this is the first time where the optical switch is observed. Why? Is there any difference between the proposed experiments compared with those developed in the previous works? Here, authors should remark what is new in their approach and why this approach let them to observe this switch. In my opinion, this is very important for a Nature Communications paper. In addition, I miss a comparison of

these results (threshold, gain...) with previous publications. For this purpose, I think is better if excitation power is specified as excitation fluency (power per area). Indeed, if the diameter of the spot is 20 μm the excitation fluency at the threshold becomes $0.53/\pi/100=170 \text{ mJ/cm}^2$, which is rather high. Are some numbers wrong here?

Our Reply: We thank the reviewer for this valuable and in-depth question. We agree with the referee that interaction-induced mode switching demonstrated here is indeed simply activated by the increase of the excitation fluency (or pump power). **However, the preferable condition that leads to this phenomenon cannot be satisfied by all types of modes.** In literatures, there are two types of lasers in perovskite micro- & nano-rods. The major type of microlaser in perovskite microrods is the Fabry-Perot lasers such as Ref. 17 (can also be seen in Fig. R4(a)). The other one is the WGM lasers that are generated in the transverse planes of microrods, see Ref. 19.

In case of Fabry-Perot lasers, below we will analytically prove that their cross-interaction coefficients χ_{12}, χ_{21} are smaller than their self-interaction coefficients χ_{11}, χ_{22} , which do not meet the requirement of interaction-induced mode switching given by Eq. (4) in the main text. To simplify our discussion, we treat them as transverse waves (ψ) described by the one-dimensional (1D) Helmholtz equation:

$$\left[\frac{d^2}{dx^2} + n^2 k^2 \right] \psi(x) = 0.$$

Here x is the axial coordinate of the microrod, n is the refractive index assumed to be uniform inside the microrod between $x = -L/2$ and $L/2$, and k is the free-space wave number. By imposing the outgoing boundary condition at both $x = -L/2$ and $L/2$, i.e.,

$$\psi(x) = \begin{cases} e^{-i(k+\frac{L}{2})x} & (x < -L/2) \\ e^{ik(x-\frac{L}{2})} & (x > L/2) \end{cases},$$

we can solve the modes inside the cavity, which are given by either

$$\psi_-(x) = b \sin(nkx) \quad \text{or} \quad \psi_+(x) = d \cos(nkx). \quad (R1)$$

In principle we need to solve for the mode-dependent wave number k and the amplitude b or d simultaneously, but since the latter will be modified using the normalization condition

$$\int_{-\frac{L}{2}}^{\frac{L}{2}} \psi^2(x) dx = L, \quad (R2)$$

we eliminate them here and focus on the solutions of k . This is done by employing the continuity boundary condition for the ratio $\frac{\psi'(x)}{\psi(x)}$, which renders the following equation for k :

$$\frac{\psi'_-(x)}{\psi_-(x)} = nk \frac{\cos\left(\frac{nkL}{2}\right)}{\sin\left(\frac{nkL}{2}\right)} = ik \quad \text{or} \quad \frac{\psi'_+(x)}{\psi_+(x)} = -nk \frac{\sin\left(\frac{nkL}{2}\right)}{\cos\left(\frac{nkL}{2}\right)} = ik.$$

or equivalently,

$$\tan\left(\frac{nkL}{2}\right) = -in \quad \text{or} \quad -\frac{i}{n}.$$

The above equation cannot be solved with a real-valued k . Instead, it has discrete solutions in

the complex plane, i.e.,

$$k_m = \frac{1}{nL} \left[m\pi - i \ln \frac{n+1}{n-1} \right],$$

where $m = 1, 3, 5, \dots$ for ψ_- and $2, 4, 6, \dots$ for ψ_+ . These k_m values are known as the (complex) resonances of the modes in a 1D Fabry-Perot cavity, while the corresponding $\psi_{\pm}(x) \equiv \psi_m(x)$ are called the resonant modes or quasi-bound modes. After the normalization specified by Eq. (R2), the constant b and d in Eq. (R1) are both given by

$$\eta_m = \left[\frac{1}{2} + \frac{i}{(1-n^2)k_m L} \right]^{-\frac{1}{2}}.$$

Figure R6. Examples of (a) an even resonant mode ($m = 4$) and (b) an odd resonant mode ($m = 5$) in a 1D Fabry-Perot cavity.

Now it is straightforward to calculate the self- and cross-interaction coefficients, and we show their numerical values in Fig. R7. It is clear that the self-interaction coefficients (diagonal bars) are greater than the cross-interaction coefficients (off-diagonal bars), and **as a result, the condition for interaction-induced mode switching are not satisfied for Fabry-Perot modes.** Even though we have only plotted their values for the 11 modes with the lowest frequency, we show below analytically that this observation holds for any resonances in a 1D Fabry-Perot cavity.

Figure R7: Self- and cross-interaction coefficients for the first 11 modes in a Fabry-Perot cavity.

From Fig. R7 we find that all self-interaction coefficients are approximately 1.5 while

the cross-interaction coefficients are roughly 1. These values can be derived analytically using the approximations

$$k_m \approx \frac{m\pi}{nL}, \quad \eta_m \approx \sqrt{2}.$$

For example, for two odd modes

$$\chi_{uv} = \frac{\eta_m^4}{L} \int_{-\frac{L}{2}}^{\frac{L}{2}} \sin(nk_u L)^2 |\sin(nk_v L)^2| dx \approx \frac{4}{L} \int_{-\frac{L}{2}}^{\frac{L}{2}} \sin(u\pi)^2 \sin(v\pi)^2 dx = 1 + \frac{\delta_{uv}}{2},$$

where $u = v$, $\delta_{uv} = 1$ for self-interaction coefficients and $u \neq v$, $\delta_{uv} = 0$ for cross-interaction coefficients. Similarly, for two even modes

$$\chi_{uv} = \frac{\eta_m^4}{L} \int_{-\frac{L}{2}}^{\frac{L}{2}} \cos(nk_u L)^2 |\cos(nk_v L)^2| dx \approx \frac{4}{L} \int_{-\frac{L}{2}}^{\frac{L}{2}} \cos(u\pi)^2 \cos(v\pi)^2 dx = 1 + \frac{\delta_{uv}}{2}$$

and the same results hold. Finally, the cross-interaction coefficients between one even mode v and one odd mode u are given by

$$\chi_{uv} = \frac{\eta_m^4}{L} \int_{-\frac{L}{2}}^{\frac{L}{2}} \sin(nk_u L)^2 |\cos(nk_v L)^2| dx \approx \frac{4}{L} \int_{-\frac{L}{2}}^{\frac{L}{2}} \sin(u\pi)^2 \cos(v\pi)^2 dx = 1,$$

$$\chi_{vu} = \frac{\eta_m^4}{L} \int_{-\frac{L}{2}}^{\frac{L}{2}} \cos(nk_v L)^2 |\sin(nk_u L)^2| dx \approx \frac{4}{L} \int_{-\frac{L}{2}}^{\frac{L}{2}} \cos(v\pi)^2 \sin(u\pi)^2 dx = 1.$$

These derivations show that indeed interaction-induced mode switching cannot take place between Fabry-Perot modes.

Compared with the Fabry-Perot cavities, the transverse WGMs are much better to realize the mode switching phenomenon. However, there are still several reasons blocking the observation of mode switching frequently. The previous studies on perovskite microrod are intending to synthesize the nanorods. In such systems, the transverse sizes are usually too small to support multiple high Q modes (see Fig. R4(b)), eliminating the possibility of mode switching between two modes. The most important reason is still the criterion of Eq. (4). As shown in Fig. R5, this criterion can only be realized with particular modes. Without the fine design, even in the large microrods, the mode switching effect is still rare due to the uncontrollability and randomness in chemical synthesis. We need to test hundreds of large perovskite rods to get several useful samples. Therefore, without a strong motivation and an extremely long time to screen the perovskite microlasers, it is hard to find the mode switching effect occasionally in the as-grown samples.

Importantly, while the production rate is low in the as-grown samples, the obtained mechanism for mode switching is quite robust and can be well reproduced. Of course, this requires the well-controlled top-down fabrication process. To verify this reproducibility, we have extended the mechanism to polymer microdisks that can be well defined with nanofabrication techniques. As shown in the supplemental information, the mode switching can be simply re-produced in a large number of samples on the same wafer. Those results

clearly show that the mode switching shall also be well reproduced in perovskite devices if the precise nanofabrication technique is further improved.

In the revised manuscript, we have added the comment in the conclusion part (para-1, page-9). “This mechanism is not limited in MAPbBr₃ perovskites. It also works well in other materials such as polymer. Importantly, with the mature top-down fabrication technique of polymer, this mode switching phenomenon can be well reproduced in a series of polymer microcavities (see Section VII in Supplementary), indicating the low production rate of switchable perovskite laser can be eventually improved with the development of top-down fabrication techniques.”

We heartfully thank the reviewer for the very careful review of the quality and threshold. In our experiments, both of the Q factors and thresholds are very close to the previous reports. The Q factors are a few thousand and the thresholds are a few $\mu\text{J}/\text{cm}^2$. The original values are directly read from the power meter and we forgot to include the reduction of optical setup and the ND filters. Following the reviewer comment, we have changed all the threshold to the pumping density in the revised manuscript.

Comment-6 Is this strategy developed previously with other materials?

Our Reply: We thank the reviewer for this careful review. This strategy has not been demonstrated in any materials before. Our experiment is the first time to illustrate this mechanism. This is also the reason that we extend the researches from perovskite laser to typical polymer laser. We want to show that the mechanism developed from our perovskite systems can also impact the conventional microlaser community.

Comment-7 I miss in the supporting information the spectra below and above the lasing threshold. Here I think it should be better to show the Y-axis in a logarithmic scale. In addition, it would be interesting to include in the supporting information the integrated intensity as a function of the excitation fluency in a log-log plot.

Our Reply: We thank the reviewer for this valuable suggestion. We have added these information in the Section IV of revised supplementary. “In this section, we will represent the detail of the lasing action. As shown in Fig. S5, we give the spectra below and above the lasing threshold, and the integrated intensity as a function of the excitation also summarized in Fig. S5.”

Figure R8: The spectra below, at, and above the lasing threshold (black, pink, and blue solid

line), and the integrated intensity as a function of the pumping density (orange circles). (Fig. S5 in supplemental information)

Comment-8 Absorption curve in Figure 1 indicates strong scattering at long wavelengths. Are results affected by this scattering? What is the size of the spot in the absorption measurements?

Our Reply: We thank the reviewer for this careful review. In the absorption measurement, the diameter of light spot is 40 μm . The increase of absorption at longer wavelength should be caused by the scattering loss of multi-perovskite microrods. The description on the absorption measurement has been included under Fig. S3 of Supplementary. “Basically, the white light is collimated and then focused by a 20 \$\times\$ objective lens onto the top surface of the sample, where the diameter of the spot is about 40 \$\mu\text{m}\$.”

Comment-9 How n and k are measured?

Our Reply: We thank the reviewer for this careful review. The n and k of perovskites are measured by Ellipsometer. In the revised manuscript (para-2, page-6), “In our calculations, the structural parameters followed the SEM image, whereas the refractive index (n) and light extinction coefficient (k) were measured by ellipsometer experimentally (see Fig. S7).”

Comment-10 I do not understand the section VII included in the supporting information. This section is even longer than the main manuscript and it is only called in the conclusions.

Our Reply: We thank the reviewer for this careful review. Section VII in the supplementary demonstrated that the proposed strategy to achieve mode switching is applicable to other materials and systems. Meanwhile, we also try to emphasize the robustness of the obtained mechanism. While the production rate is low, it is mainly because that the synthesized samples have random sizes. Once the cavity size and shapes can be precisely controlled with the top-down nanofabrication technique, the obtained mechanism can be easily reproduced in many samples on the same wafer. We hope this information can impact the microlaser researches in other material systems.

Comment-11 I found misprints in line 70 "confim" by "confirms", 109 mechanism by mechanisms, line 169 "spectrum" by spectrum.

Our Reply: We thank the reviewer for this careful review. We have checked through the manuscript carefully and corrected the typos, misspellings and grammatical errors.

Comment-12 Reference 39 is missing.

Our Reply: We thank the reviewer for this careful review. The missing reference has been added in the revised manuscript, and adjust to reference 36. “36. Zhang N. et al. Postsynthetic and Selective Control of Lead Halide Perovskite Microlasers, *Journal of Physical Chemistry Letters*, 7, 3886-3891 (2016).”

Comment-13 Line 56. Include a reference about cross gain saturation observed with other materials, and explain what it is.

Our Reply: We thank the reviewer for this careful review. We would like to thank the referee for these good suggestions. A reference on cross gain saturation has been included in Line 56

[Türeci, H. E., Ge, L., Rotter, S. & Stone, A. D. Strong interactions in multimode random lasers. *Science* **320**, 643 (2008). Ge, L., Malik, O. & Türeci, H. E. Enhancement of laser power-efficiency by control of spatial hole burning interactions. *Nature Photonics* **8**, 871-875 (2014).], and the sentence has been rewritten as the following: “In this research, we explore nonlinear modal interactions in perovskite microlasers and demonstrate their impacts on ultrafast mode switching, especially with cross gain saturation^{46,47} where the intensity of one lasing mode reduces the available gain for all other modes in the same system.”

Comment-14 I do not understand why the inset of Figure 2d is referred in Line 142. Anyway, this inset is not commented within the Figure (lines 95-104).

Our Reply: We thank the reviewer for this valuable suggestion. In the revised manuscript, the description on the inset of Figure 2(d) has been added in para-2, page-4. “The inset of Fig. 2(d) shows the polarization of these two WG modes, which are both transverse electrically (TE) polarized with dominant electric field perpendicular to the light propagation direction in the cross-sectional plane.”

Comment-15 Line 149. Are cross-section interaction coefficients normalized magnitudes? or are units missed?

Our Reply: We thank the reviewer for this careful review. We thank the reviewer for this excellent question. Here the field distributions $u_\mu(\vec{r})$ and $u_\nu(\vec{r})$ are normalized and dimensionless, and hence the cross-interaction coefficients defined in Line 149 is also dimensionless. We have clarified this point by revising the sentence below Eq. (2) to “... \$u_\mu(\vec{r})\$ is the normalized and dimensionless field distribution of mode- \$\mu\$ in the cavity.” We also corrected a related typo in Eq. (S7) of the SI, which is now consistent with the rest of the manuscript.

Reply to Reviewer #2:

We would like to thank the reviewer for the valuable suggestions and recommendation for publication in Nature Communications after a minor revision. Based on the comments, we have carefully revised our manuscript and the manuscript quality has been significantly improved.

Comment-1. The references in the introduction section should be carefully checked, e.g. though the introduction mainly focuses on MAPbX₃, some works on all inorganic perovskites have been cited; on Line-41, Page-2, the references 28-33 are not about photonic crystals; the reference 39 is missing in the reference list.

Our Reply: We thank the reviewer for this valuable suggestion. The references have been checked and corrected carefully. Inappropriate references have been deleted or replaced with proper references. The original references “28-33” should be about distributed feedback structures and the missing reference “39” in the original manuscript has been added. “36. Zhang N. et al. Postsynthetic and Selective Control of Lead Halide Perovskite Microlasers, Journal of Physical Chemistry Letters, 7, 3886-3891 (2016).”

Comment-2. In Line-38, Page-2, the record values on threshold and Q factor of perovskite microlasers cannot be found in the reference. The authors should very accurate information or citation.

Our Reply: We thank the reviewer for this careful review. The record threshold and Q factor of perovskite microlasers have been corrected in the revised manuscript in para-1, page-2. “The threshold and quality (Q) factors of perovskite microlasers have been improved to 220 nJ/cm² ¹⁷ and 1×10⁴ ¹², respectively.”

Comment-3. In Figure 1, authors mainly discuss the crystalline quality and optical properties of perovskites, which are not directly relevant to the topic of the manuscript. Thus I suggest that authors add them into the supporting information.

Our Reply: We thank the reviewer for this valuable suggestion. We agree with the reviewer that some detail analysis of structural information can be placed in the supplemental information. Following the reviewer’s suggestion, we have compressed the contents of this part (para-2, page-3) in the revised manuscript. “The single crystal nature of synthesized perovskite microrods were determined by the following X-ray diffraction (XRD) spectrum and high-resolution transmission electron microscopy (HRTEM) investigation (see Fig. S2). Figure 1(b) shows the absorption and photoluminescence spectra of MAPbBr₃ perovskites microrods (see **methods**, Fig. S3 and Fig. S4). A clear bandedge can be seen at ~ 2.32 eV, consistent with the previous reports well¹².” As shown in Fig. R9 below, the corresponding contents in Fig. 1 have also been modified.

The structural information of MAPbBr₃ perovskite microrod have been moved to Fig. S2 in the supporting information in Section I. “In the main text, in order to characterize the material properties of single crystal perovskite microrod, we use the high-resolution transmission electron microscopy (HRTEM) and the fast Fourier transform (FFT) pattern.”

associated with the XRD spectrum, clearly confirms the single crystal nature of the synthesized MAPbBr₃ microrods. Fig. S2(b) is the X-ray diffraction (XRD) spectrum of the synthesized nanorods. Four sharp peaks appear at 15°, 30°, 46°, and 63°, which can be indexed to the (001), (002), (003), and (004) crystal planes of cubic phase of MAPbBr₃ perovskites.”

Figure R9: The synthesized MAPbBr₃ perovskite microrods. (a) The top-view SEM image of MAPbBr₃ microrods. (b) The absorption (green line) and photoluminescence (orange line) of MAPbBr₃ perovskite microrods. (Shown as Fig. 1 in the main manuscript.)

Figure R10. (a) The HRTEM image and corresponding FFT pattern of MAPbBr₃ perovskite. (b) The XRD spectrum of synthesized perovskite microrods. (Shown as Fig. S2 in the supplemental information.)

Comment-4. The description on the excitation is not proper: pumping density should be changed to pumping energy, considering that joules are used in the manuscript to describe the excitation.

Our Reply: We thank the reviewer for this valuable suggestion. In the revised manuscript, the descriptions on excitation fluence have been changed to pumping density with unit of $\mu\text{J}/\text{cm}^2$.

Comment-5. In Figure 2c, the intensity of mode 2 slightly drops when the excitation energy is above 0.8 μJ . What is the cause of the drop? It seems that the Auger effect starts to play an important role. The author should discuss this and give a possible way to solve this effect.

Our Reply: We thank the reviewer for this careful review and valuable comment. The reviewer is absolutely right that Auger recombination began to affect the output power at high pumping fluence. As a result of multiple-particle loss (Auger nonradiative recombination), the perovskite microlaser exhibits a flat or even negative power slope at high pumping density (Zhang, C. et al. *Adv. Opt. Mater.* 4, 2057-2062 (2016)). Fortunately, the perovskite

microlasers can eventually overcome this limitation. As shown in Ref. 48 (Zhang, C. et al. Adv. Opt. Mater. 4, 2057-2062 (2016)), the Auger effect can be significantly eliminated by simply covering a few layer graphene. In the revised manuscript, the discussion on the intensity drop at high pumping density in Fig. 2(c) has been added (para-2, page-4). “It is worth noting that the emission intensity of mode-2 shows a slight drop with further increase of pumping density. This caused by Auger recombination at high pumping fluence. This limitation can be reduced with additional technique such as covering the sample with few-layer graphene⁴⁸”

Comment-6. Do the two competing modes in Figure 2b have the same polarization? The polarization shown in the inset of Figure 2d has not been described with detail information.

Our Reply: These two WG modes have the same polarization. In the revised manuscript, the description on the inset of Figure 2(d) has been added in para-2, page-4. “The inset of Fig. 2(d) shows the polarization of these two WG modes, which are both transverse electrically (TE) polarized with dominant electric field perpendicular to the light propagation direction in the cross-sectional plane.” and the detail of the polarization is shown in Figure R11.

Figure R11: The polarization of emitted light of the perovskite microrod.

Comment-7. Typos and grammar errors should be checked through the whole manuscript, e.g. on Line-21, Page-1, “reversible” should be “reversible”; on Line-22&23, Page-1, there are two redundant hyphens.

Our Reply: We thank the reviewer for this careful review. We have checked through the manuscript carefully and corrected the typos and grammar errors.

REVIEWERS' COMMENTS:

Reviewer #1 (Remarks to the Author):

Now authors have addressed all my comments and I think the paper is ready for publication. I have only found the following minor issues:

1. Line 23. Replace "doesn't" by "does not"
2. Line 120. Replace "In literatures" by "In the literature".
3. Check the use of the article the along the text.
4. I think is necessary to explain the conditions to observe these modes. I would recommend the authors to include a section in the supporting information with their response to my comments 4 and 5, and a brief summary (1-2 sentences) in the discussion section.

Congratulations for this nice work,
Isaac Suárez

Reviewer #2 (Remarks to the Author):

I thought that the authors did a great job to revise the manuscript. I am satisfied with the revisions, and recommend the publication in its current form.

Reply to Reviewer #1:

We thank the reviewer for the important suggestions and valuable comments. Based on the reviewer's comment, we have carefully revised our manuscript again.

Comment-1 Line 23. Replace "doesn't" by "does not"

Our Reply: We thank the reviewer for this careful review. We have correct this error in our revised manuscript.

Comment-2 Line 120. Replace "In literatures" by "In the literature".

Our Reply: We thank the reviewer for this careful review. We have checked through the manuscript carefully and corrected the typos, misspellings and grammatical errors.

Comment-3 Check the use of the article the along the text.

Our Reply: We thank the reviewer for this valuable suggestion. The articles have been checked and corrected carefully.

Comment-4 I think is necessary to explain the conditions to observe these modes. I would recommend the authors to include a section in the supporting information with their response to my comments 4 and 5, and a brief summary (1-2 sentences) in the discussion section.

Our Reply: We thank the reviewer for this valuable suggestion. We have add these discussions as "Supplementary Note 4" in the revised supporting information, and we also added a brief summary in para-1,page 8 of the revised manuscript. "In experiments, the preferable condition that leads to this phenomenon cannot be satisfied by all types of modes, and based on this strict requirement of the criterion and the random sizes of as-grown, mode switching is not ubiquitous in the as-grown perovskite microlasers (detailed see Supplementary Note 4). This can be solved by high-quality top-down nanofabrication technique to precisely fabricate the designed cavities (see Supplementary Note 8)."

Reply to Reviewer #2:

Comment: I thought that the authors did a great job to revise the manuscript. I am satisfied with the revisions, and recommend the publication in its current form.

Our Reply: We would like to thank the reviewer for the valuable suggestions. We also appreciate that the reviewer recognize the value of our research and recommended for acceptance by Nature Communications.